REGISTERED REPORT PROTOCOL

# Artificial intelligence in human resource development: An umbrella review protocol

**Sangok Yoo** *, **Kim Nimon, Sanket Ramchandra Patole**

Human Resource Development, The University of Texas at Tyler, Tyler, Texas, United States of America

* syoo@uttyler.edu

## Abstract

The recent surge in artificial intelligence (AI) has significantly transformed work dynamics, particularly in human resource development (HRD) and related domains. Scholars, recognizing the significant potential of AI in HRD functions and processes, have contributed to the growing body of literature reviews on AI in HRD and related domains. Despite the valuable insights provided by these individual reviews, the challenge of collectively interpreting them within the HRD domain remains unresolved. This protocol outlines the methodology for an umbrella review aiming to systematically synthesize existing reviews on AI in HRD. The review seeks to address key research questions regarding AI's contributions to HRD functions and processes, as well as the opportunities and threats associated with its implementation by employing a technology-aided systematic approach. The coding framework will be used to synthesize the contents of the selected systematic reviews such as their search strategies, data synthesis approaches, and HRD-related findings. The results of this umbrella review are expected to provide insights for HRD scholars and practitioners, promoting continuous improvement in AI-driven HRD initiatives. This protocol is preregistered on the Open Science Framework (https://doi.org/10.17605/OSF.IO/Z8NM6) on May 27, 2024.

## Introduction

Artificial intelligence (AI) refers to the ability of machines to perform near or human-like functions, such as learning, interaction, and problem-solving, encompassing the culmination of computers, computer-related technologies, machines, and information communication technology innovations and developments, giving computers the ability to perform [1, 2]. The AI market is anticipated to reach a $407 billion by 2027, indicating substantial growth from its estimated revenue of $86.9 billion in 2022. This surge is projected to make a 21% net contribution to the United States GDP by 2030, highlighting the profound impact of AI on economic growth [3]. Reasonably, a considerable 64% of businesses believe artificial intelligence will enhance their overall productivity [3]. Furthermore, according to an annual McKinsey Global Survey conducted in mid-April 2023, generative AI (Gen AI) has captured significant attention across the business landscape. Individuals from various regions, industries, and seniority levels are incorporating Gen AI into their professional and personal activities in their workplaces [4].

**Data Availability Statement:** All supplementary files are available in an open-access repository: https://osf.io/af6d7/.

**Funding:** The author(s) received no specific funding for this work.

**Competing interests:** The authors have declared that no competing interests exist.

The recent proliferation of AI has dramatically changed the way we work [2, 5]. In the field of human resource development (HRD) and related areas, the integration of AI presents opportunities to optimize talent acquisition, streamline learning and development initiatives, and enhance the strategic values of HRD in the workplace [5, 6]. The far-reaching impact of AI underscores the need for a nuanced understanding of its role in HRD functions.

In academia, a burgeoning interest in AI in the workplace is evident through the growing body of research, leading to a surge of literature reviews focused on AI in HRD and related areas (e.g., [2, 5, 7]). For example, [5] conducted a critical review of the literature on AI and its impact on workplace outcomes, specifically within HR functions. [6] delved into the literature on AI applications, with a particular emphasis on the learning and development function. Despite the valuable contributions of these endeavors, the question of how these individual reviews can be collectively interpreted within the field of HRD remains unanswered.

To attain a comprehensive understanding of the rapidly expanding knowledge base, there is a need to systematically synthesize existing reviews on AI in HRD and related areas. An umbrella review, representing the highest level of evidence, offers a comprehensive overview of existing systematic reviews in a specific field. It enables scholars to compare the findings of systematic reviews relevant to a specific review question [8, 9].

Hence, the proposed review outlined in this protocol aims to unveil patterns, trends, and gaps in the current understanding of AI in HRD literature. Additionally, we expect that this umbrella review will provide HRD scholars and practitioners with insights into the evolving concepts and practices associated with AI in HRD, thereby promoting continuous improvement in AI-driven HRD initiatives. The key research questions to be addressed in our umbrella review are:

*RQ1*: *How does AI contribute to HRD functions and processes*?

*RQ2*: *What are the opportunities and threats of implementing AI in HRD*?

In pursuit of the objective, this protocol proposes a technology-aided umbrella review process to synthesize systematic literature reviews on AI in the field of HRD and related areas. This systematic approach is designed to alleviate subjectivity in the review process, including the selection of search terms, thereby enhancing the rigor and objectivity of this umbrella review.

## Materials and methods

### Design and setting of the study

This technology-aided umbrella review protocol adheres to the guidelines of PRISMA-P (Preferred reporting items for systematic review and meta-analysis protocols), serving as a guide for planning and documenting review methods [10, 11]. The completed PRISMA-P checklist to confirm essential and minimum components of a systematic review is available in the S1 File. To achieve a comprehensive understanding of AI implementation in HRD, this protocol is designed to systematically incorporate existing systematic literature reviews on AI in HRD and related areas, mitigating subjective decision-making during review conduct [10]. This protocol is pre-registered on the Open Science Framework (OSF): https://doi.org/10.17605/OSF.IO/Z8NM6. In the main research using this protocol, we plan to incorporate guidelines from the updated PRISMA 2020 statement to ensure comprehensive reporting of our umbrella review [12].

## Database and data management

A structured search will be conducted in the Scopus and Web of Science databases, selected for their relevance to the field of study and comprehensive coverage. The review process, encompassing screening, will be coordinated utilizing Rayyan to ensure a systematic and efficient workflow [13].

## Search strategy

Keywords to create a comprehensive search string that will be used to search systematic reviews for this umbrella review were collected. Table 1 describes the final search sub-strings of each component. AI-, HRD-, and SLR-related strings include search terms combined using the Boolean operator OR. In the final search string, the Boolean operator AND will be used to combine the three sub-strings. As our umbrella review aims to synthesize existing systematic literature reviews, the SLR-related string includes one search term that narrows the scope of our project. The specific search term identification strategy and term matching details are illustrated in the supplementary files (S2 and S3 Files).

## Screening process

We will employ a two-stage screening strategy. First, the relevance of each article will be evaluated based on its title and abstract. Articles that meet the exclusion criteria will be excluded. The second stage will evaluate the relevance of articles based on full texts using the inclusion and exclusion criteria. The screening process will be coordinated using Rayyan.

## Eligibility criteria

To uphold consistency and reproducibility in the screening process among coders, the inclusion and exclusion criteria are established. First, eligible studies are systemic literature reviews

**Table 1. Final search sub-strings of each component [a].**

| AI-related string | HRD-related string | SLR-related string |
|---|---|---|
| "AI" OR algorithm* OR automation OR "artificial intelligen*" OR "artificial-intelligen*" OR "augmented reality" OR "autonomous agent*" OR bayesian* OR block-chain OR blockchain OR "business intelligence" OR chat* OR "cloud computing" OR cloud-computing OR "collaborative intelligence" OR "collective intelligence" OR "competitive intelligence" OR "complex network*" OR computation* OR computer* OR "conversational agent*" OR "deep learning" OR "digital transformation" OR "digital twin*" OR "expert system*" OR "face recognition" OR "facial recognition" OR fuzzy* OR "human-agent interaction*" OR "human-computer interaction*" OR "human-robot interaction*" OR "robot-human interaction*" OR "human computer interaction*" OR "human machine interaction*" OR "human robot interaction*" OR "image recognition" OR "industry 4.0" OR "industry 5.0" OR "intelligent agent*" OR "internet of thing*" OR IoT OR "language processing" OR "large language model*" OR LLM OR "machine intelligence" OR "machine learning" OR ML OR "natural language*" OR "neural network*" OR neural-network* OR NLP OR "pattern recognition" OR "random forest*" OR "recommendation engine*" OR "remote monitoring" OR "remote sensing" OR robot* OR "smart device*" OR "society 5.0" OR "speech recognition" OR "support vector* machine*" OR SVM OR technolog* OR "text mining" OR "text processing" OR virtual* OR "wearable sensor*" OR "wireless sensor network*" | "action learning" OR "career development" OR "CD" OR "change management" OR coach* OR "corporate social responsibility" OR creativity OR "CSR" OR cultur* OR diversity OR e-hrm OR "employ* experience*" OR "employ* relation*" OR "employee analytic*" OR "employee performance" OR engagement OR "environmental, social, and corporate governance" OR "ESG" OR "future of work*" OR "HR" OR "HR analytic*" OR "HRD" OR "HRM" OR human-capital OR "human capital" OR human-resource* OR "human resource*" OR innovation OR job* OR knowledge* OR leader* OR learn OR "OC" OR "OD" OR "organization* change" OR "organization* development" OR "organization* performance" OR "people analytic*" OR "performance appraisal" OR "performance assessment" OR "professional development" OR retention OR skill OR succession OR "talent analytic*" OR "talent development" OR "talent management" OR "task performance" OR team* OR train* OR turnover OR workforce OR workplace* | "systematic literature review" |

[a] The Boolean operator AND will be used to combine the three components.

specifically focused on AI in the field of HRD and related areas. This inclusion criterion aims to contribute to the synthesis of high-quality evidence and insights derived from rigorous research methodologies. The initial search will be confined to peer-reviewed journal articles and conference proceedings written in English and published from 1995 onwards, aligning with the search practices in previous literature reviews on AI (e.g., [5–7]).

Regarding the exclusion criteria, first, studies that do not explicitly explore AI-related technology will be excluded, ensuring a targeted exploration of the subject matter. Second, studies unrelated to a workplace setting will be excluded, as this umbrella review is specifically tailored to the application of AI in the workplace. Third, non-systemic literature reviews, which lack a structured and systematic approach, will also be excluded to maintain the methodological rigor of the review. Fourth, as this umbrella review specifically targets systemic literature reviews, studies employing meta-analysis as the primary research methodology will not be considered for inclusion. Finally, as explained in the inclusion criteria, book chapters and non-referred articles will be excluded to maintain the scholarly standard and reliability of the information under consideration.

## Data extraction

We will use Rayyan to extract data. Extraction fields will be set up with the relevant information from the studies, and Rayyan's tagging and coding features will be used to categorize and organize the extracted data. Disagreements will be discussed and resolved using Rayyan's conflict resolution feature. The extraction fields for recording the finally selected systematic review studies will include:

- Review article information

  - Full study citation

  - The number of citations

  - Title, abstract, and keywords

  - Publication outlet (e.g., journal) and year

- Details of the search strategy used in the study

  - Database, journal types, research context, scope

  - Timeframe

  - Search terms and string(s)

  - Scope of AI-related technologies (e.g., AI, machine learning, large language model)

  - Scope of HRD-related functions (e.g., training & development, organizational development)

- Details of the data analysis strategy in the study

  - Analysis approaches (e.g., bibliometrics, contents analysis, topic modeling, clustering)

- HRD-related findings in the study

  - HRD-related areas in which AI applies to

  - The benefits and possibility of AI adoption in HRD functions

  - The enablers and obstacles of AI adoption in HRD functions

- Contributing factors to the effectiveness of AI-based HRD practices
- Other key contents/findings of the study (e.g., Future research directions)

### Data synthesis

Thematic coding will be a crucial part of this umbrella review, focusing on discerning patterns in the implementation of AI within HRD. By employing an HRD framework, the goal of the thematic coding is to systematically categorize and analyze relevant literature to identify recurrent themes and trends in AI adoption across various HRD contexts. Furthermore, thematic coding facilitates the identification of key opportunities and challenges associated with AI implementation in HRD. The synthesis can highlight common issues faced by organizations integrating AI into HRD practices and, conversely, showcase successful strategies and innovative approaches. Ultimately, the thematic coding approach provides a comprehensive understanding of the current state of AI in HRD and sets the stage for suggesting future research directions and practical recommendations to enhance AI-driven HRD initiatives.

In addition to thematic coding, the data synthesis plan incorporates descriptive statistics. Descriptive statistics involves quantifying the occurrence of specific themes or concepts related to AI implementation in HRD across the selected systematic literature reviews. Specifically, frequency analysis helps to identify the prevalence of certain trends, challenges, or opportunities and visualization techniques can be employed to present these findings in a clear and accessible manner. R will be utilized for statistical analysis and visualization. We plan to use the base package [14] for statistical analysis and ggplot2 [15] for visualization.

### Conclusions

This protocol will guide an umbrella review process to synthesize existing systematic reviews on AI in HRD. This umbrella review aims to explore the intersection of AI and HRD using existing reviews in the field of HRD and related areas. The anticipated outcomes of this umbrella review are intended to unveil patterns, opportunities, and threats of AI implementation in HRD. They will provide insights into AI-driven HRD initiatives. All data and analyses will be placed in an open-access repository, and the URL will be provided in the final manuscript.

Despite the expected contributions of this project, several limitations should be discussed. First, the protocol's reliance on systematic literature reviews may introduce a potential bias, as certain valuable perspectives from non-systematic reviews or other types of reviews may be overlooked. Second, the scope of the review is contingent upon the availability of relevant literature published in English from 1995 onwards; this temporal and linguistic restriction may exclude valuable insights from non-English publications or earlier works that could contribute to a more nuanced understanding of the historical development of AI in HRD. Lastly, it should be mentioned that as AI-related technology is evolving rapidly future updates to this umbrella review will be necessary to ensure that it includes the most updated trends and practices.

### Supporting information

**S1 File. PRISMA-P checklist (https://osf.io/2935t).**
(DOCX)

**S2 File. Search term identification strategy (https://osf.io/vgck2).**
(DOCX)

**S3 File. VosViewer keywords and search terms matching (https://osf.io/nxc7v).** *****Note**: All supplementary files are available in an open-access repository: https://osf.io/af6d7/.
(XLSX)

## Author Contributions

**Conceptualization:** Sangok Yoo, Kim Nimon, Sanket Ramchandra Patole.

**Data curation:** Sangok Yoo, Kim Nimon, Sanket Ramchandra Patole.

**Methodology:** Sangok Yoo, Kim Nimon.

**Project administration:** Sangok Yoo.

**Software:** Kim Nimon.

**Supervision:** Sangok Yoo, Kim Nimon.

**Validation:** Sangok Yoo, Kim Nimon, Sanket Ramchandra Patole.

**Writing – original draft:** Sangok Yoo.

**Writing – review & editing:** Sangok Yoo, Kim Nimon, Sanket Ramchandra Patole.

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
