## [Decision Letter · Decision Letter 0]

9 May 2024

PONE-D-24-05416Artificial intelligence in Human Resource Development:  An umbrella review protocolPLOS ONE

Dear Dr. Yoo,

Thank you for submitting your manuscript to PLOS ONE. After careful consideration, we feel that it has merit but does not fully meet PLOS ONE’s publication criteria as it currently stands. Therefore, we invite you to submit a revised version of the manuscript that addresses the points raised during the review process.

We look forward to receiving your revised manuscript.

Kind regards,

Antonio L. García-Izquierdo, Ph.D.

Academic Editor

PLOS ONE

Journal Requirements:

Additional Editor Comments :

Dear Doctor Yoo,

I finally received the feedback from our reviewers. Their comments pinpoint critical weaknesses that should be addressed. If you can manage the following comments, you can significantly improve the quality of your manuscript. Be aware that my decision should not be assumed as a preliminary acceptance of your research. Given the topic of your work amid the emergence of artificial intelligence, the manuscript has some merits, but the document demands major revisions following suggestions provided as follows:

Reviewer #1

Here are my comments on this manuscript to improve its clarity:

• I recommend citing and reviewing the updated PRISMA document, "The PRISMA 2020 statement: an updated guideline for reporting systematic reviews" published in BMJ 2021; 372 doi: <https: 10.1136="" bmj.n71="" doi.org="">.

• Consider supplementing the database search with AI tools such as Perplexity and Consensus.

• I was unable to access the supplementary files, so I cannot provide comments on them.

• The authors mention that they will use a different program than Covidence for information extraction since it only allows two coders. Please specify which program will be used.

• It is unclear which descriptive statistics will be used and which programming language will be utilized for this purpose. Is R being used?

• Additionally, will inferential statistics be employed in the study?

• Data and all analyses should be placed in a repository and the URL provided in the manuscript.

Reviewer #2

In their work, the authors explored different ways to define research terms based on a initial search with key terms such as artificial intelligence, large language model, machine learning and human resource, from which they extracted more specific terms and also included some trendy terms related with analytics, and after that, the relevance of the resulting terms were validated using the VOSViewer software, to finally propose a search string to developing the review in databases such as Scopus or WoS.

Unfortunately, the proposed protocol describes a search process in which the authors made some decisions that made the protocol very specific, when they included, for instance, terms related with analytics, only because they are trendy.

Another aspect affecting the replicability of the protocol is that the authors do not list, in the references, the 16 studies from which extract AI-related and HDR search terms.

For these reasons, from my point of view, the protocol presented here is not replicable.

Reviewers' comments:

Reviewer's Responses to Questions</https:>

**Comments to the Author**

1. Does the manuscript provide a valid rationale for the proposed study, with clearly identified and justified research questions?

Reviewer #1: Yes

Reviewer #2: Yes

2. Is the protocol technically sound and planned in a manner that will lead to a meaningful outcome and allow testing the stated hypotheses?

Reviewer #1: Partly

Reviewer #2: Partly

3. Is the methodology feasible and described in sufficient detail to allow the work to be replicable?

Reviewer #1: No

Reviewer #2: No

4. Have the authors described where all data underlying the findings will be made available when the study is complete?

Reviewer #1: No

Reviewer #2: No

5. Is the manuscript presented in an intelligible fashion and written in standard English?

Reviewer #1: Yes

Reviewer #2: Yes

6. Review Comments to the Author

You may also provide optional suggestions and comments to authors that they might find helpful in planning their study.

Reviewer #1: Here are my comments on this manuscript to improve its clarity:

• I recommend citing and reviewing the updated PRISMA document, "The PRISMA 2020 statement: an updated guideline for reporting systematic reviews" published in BMJ 2021; 372 doi: <https: 10.1136="" bmj.n71="" doi.org="">.

• Consider supplementing the database search with AI tools such as Perplexity and Consensus.

• I was unable to access the supplementary files, so I cannot provide comments on them.

• The authors mention that they will use a different program than Covidence for information extraction since it only allows two coders. Please specify which program will be used.

• It is unclear which descriptive statistics will be used and which programming language will be utilized for this purpose. Is R being used?

• Additionally, will inferential statistics be employed in the study?

• Data and all analyses should be placed in a repository and the URL provided in the manuscript.

Best regards,

FMR</https:>

Reviewer #2: After reading the manuscript title “Artificial intelligence in Human Resource Development: An umbrella review protocol” in which the authors present a protocol to development a umbrella review about the use of AI in HRD focused only in review articles. In their work, the authors exploring different ways to define research terms based on a initial search with key terms such as artificial intelligence, large language model, machine learning and human resource, from which they extract more specific terms and also include some trendy terms related with analytics, and after that, the relevance of the resulting terms were validated using the VOSViewer software, to finally proposed a search string to developing the review in databases such as Scopus or WoS.

Unfortunately, the proposed protocol describes a search process in which the authors made some decisions that made the protocol very specific, when they included, for instance, terms related with analytics, only because they are trendy.

Another aspect affecting the replicability of the protocol is that the authors do not list, in the references, the 16 studies from which extract AI-related and HDR search terms.

For these reasons, from my point of view, the protocol presented here is not replicable.

7. PLOS authors have the option to publish the peer review history of their article (what does this mean?). If published, this will include your full peer review and any attached files.

Reviewer #1: **Yes: **Fernando Marmolejo-Ramos

Reviewer #2: No

---

## [Author Response · Author response to Decision Letter 0]

28 May 2024

We greatly appreciate the valuable comments. They prompted us to revise our manuscript thoroughly, with the hope of enhancing its overall quality. Please see the response table attached. We have provided our point-by-point explanations addressing each comment by specific areas of focus.

---

## [Editor Report · Decision Letter 1]

4 Jul 2024

PONE-D-24-05416R1Artificial intelligence in Human Resource Development:  An umbrella review protocolPLOS ONE

Dear Dr. Yoo,

Thank you for submitting your manuscript to PLOS ONE. After careful consideration, we feel that it has merit but does not fully meet PLOS ONE’s publication criteria as it currently stands. Therefore, we invite you to submit a revised version of the manuscript that addresses the points raised during the review process. Please submit your revised manuscript by Aug 18 2024 11:59PM. If you will need more time than this to complete your revisions, please reply to this message or contact the journal office at plosone@plos.org. Please include the following items when submitting your revised manuscript:A rebuttal letter that responds to each point raised by the academic editor and reviewer(s). You should upload this letter as a separate file labeled 'Response to Reviewers'.A marked-up copy of your manuscript that highlights changes made to the original version. You should upload this as a separate file labeled 'Revised Manuscript with Track Changes'.An unmarked version of your revised paper without tracked changes. You should upload this as a separate file labeled 'Manuscript'.If applicable, we recommend that you deposit your laboratory protocols in protocols.io to enhance the reproducibility of your results. Protocols.io assigns your protocol its own identifier (DOI) so that it can be cited independently in the future. For instructions see: https://journals.plos.org/plosone/s/submission-guidelines#loc-laboratory-protocols. Additionally, PLOS ONE offers an option for publishing peer-reviewed Lab Protocol articles, which describe protocols hosted on protocols.io. Read more information on sharing protocols at https://plos.org/protocols?utm_medium=editorial-email&utm_source=authorletters&utm_campaign=protocols.

We look forward to receiving your revised manuscript.

Kind regards,

Juan C Correa

Academic Editor

PLOS ONE

Journal Requirements:

Additional Editor Comments:

Dear Dr. Yoo,

I have read the most recent version of the manuscript and I think it addresses the reviewers' comments. Based on reviewers' comments and my own reading of your manuscript, I think your paper have shown a reasonable review protocol. There is, however, a final minor detail regarding your manuscript that will be of great value for our readers. In the last version of the manuscript, you mentioned: "R will be utilized for statistical analysis and visualization." This statement is quite generic and provides no clear guidance. Please be specific regarding the libraries or packages you are going to use. For example, if you plan to do some data visualizations, be aware you can use standard libraries such as R base, or more specialized libraries such as "ggplot2" or "ggstatsplot." Likewise, some analyses can take advantage of using network data such as "igraph" or "statnet" (for authors co-citations analysis) and/or textual data by using packages such as "quanteda" or "tidytext." Be aware that some of these data analysis and data visualizations can be easily achieved with "bibliometrix" and its shiny app "biblioshiny." A short statement illustrating these intended resources can be achieved by including one or two sentences with some of these details. Once these inclusions are addressed, the manuscript can be accepted.

---

## [Author Response · Author response to Decision Letter 1]

9 Jul 2024

You can also find our responses to the comments in the attached response letter.

Additional Editor Comments: I have read the most recent version of the manuscript and I think it addresses the reviewers' comments. Based on reviewers' comments and my own reading of your manuscript, I think your paper have shown a reasonable review protocol. There is, however, a final minor detail regarding your manuscript that will be of great value for our readers. In the last version of the manuscript, you mentioned: "R will be utilized for statistical analysis and visualization." This statement is quite generic and provides no clear guidance. Please be specific regarding the libraries or packages you are going to use. For example, if you plan to do some data visualizations, be aware you can use standard libraries such as R base, or more specialized libraries such as "ggplot2" or "ggstatsplot." Likewise, some analyses can take advantage of using network data such as "igraph" or "statnet" (for authors co-citations analysis) and/or textual data by using packages such as "quanteda" or "tidytext." Be aware that some of these data analysis and data visualizations can be easily achieved with "bibliometrix" and its shiny app "biblioshiny." A short statement illustrating these intended resources can be achieved by including one or two sentences with some of these details. Once these inclusions are addressed, the manuscript can be accepted.

Responses: Thank you for the suggestion to specify the packages we plan to use in the analysis stage. As this study involves only descriptive statistics and visualizations, we plan to use R base and ggplot2 for this project. We’ve added a sentence to specify the packages on page 10 of the main protocol: 

“R will be utilized for statistical analysis and visualization. We plan to use the base package (R Core Team, 2024) for statistical analysis and ggplot2 (Wickham, 2016) for visualization” (p.10).

It should be noted that, as stated in the data synthesis section, we plan to primarily use thematic coding by employing an HRD framework. While we will definitely consider using SNA, text analysis, and bibliometrics for future potential projects, we do not plan to use them in the current project using this protocol.

Journal Requirements: Please review your reference list to ensure that it is complete and correct. If you have cited papers that have been retracted, please include the rationale for doing so in the manuscript text, or remove these references and replace them with relevant current references. Any changes to the reference list should be mentioned in the rebuttal letter that accompanies your revised manuscript. If you need to cite a retracted article, indicate the article’s retracted status in the References list and also include a citation and full reference for the retraction notice.

Responses: We have reviewed the reference lists in the main protocol and the supplementary file (S2_File.docx) and confirmed that there are no retracted articles. Therefore, no changes are needed to the reference lists.

---

## [Editor Report · Decision Letter 2]

26 Aug 2024

Artificial intelligence in Human Resource Development:  An umbrella review protocol

PONE-D-24-05416R2

Dear Dr. Yoo,

We’re pleased to inform you that your manuscript has been judged scientifically suitable for publication and will be formally accepted for publication once it meets all outstanding technical requirements.

Kind regards,

Juan Correa

Academic Editor

PLOS ONE
---

## [Editor Report · Acceptance letter]

29 Aug 2024

PONE-D-24-05416R2 

PLOS ONE

Dear Dr. Yoo, 

I'm pleased to inform you that your manuscript has been deemed suitable for publication in PLOS ONE. Congratulations! Your manuscript is now being handed over to our production team.

Kind regards, 

on behalf of

Dr. Juan Correa 

Academic Editor

PLOS ONE